# The Upper Nasal Space: Option for Systemic Drug Delivery, Mucosal Vaccines and “Nose-to-Brain”

**DOI:** 10.3390/pharmaceutics15061720

**Published:** 2023-06-13

**Authors:** Stephen B. Shrewsbury

**Affiliations:** Formerly of Impel Pharmaceuticals, 201 Elliott Ave West, Seattle, WA 98119, USA; steve.shrewsbury@me.com; Tel.: +1-(415)-250-1169

**Keywords:** nasal delivery, intranasal delivery, upper nasal space, olfactory epithelium, blood-brain barrier (BBB), central nervous system (CNS), ViaNase^®^, OptiNose^®^, Precision Olfactory Delivery (POD^®^), nose-to-brain (N2B)

## Abstract

Sino-nasal disease is appropriately treated with topical treatment, where the nasal mucosa acts as a barrier to systemic absorption. Non-invasive nasal delivery of drugs has produced some small molecule products with good bioavailability. With the recent COVID pandemic and the need for nasal mucosal immunity becoming more appreciated, more interest has become focused on the nasal cavity for vaccine delivery. In parallel, it has been recognized that drug delivery to different parts of the nose can have different results and for “nose-to-brain” delivery, deposition on the olfactory epithelium of the upper nasal space is desirable. Here the non-motile cilia and reduced mucociliary clearance lead to longer residence time that permits enhanced absorption, either into the systemic circulation or directly into the CNS. Many of the developments in nasal delivery have been to add bioadhesives and absorption/permeation enhancers, creating more complicated formulations and development pathways, but other projects have shown that the delivery device itself may allow more differential targeting of the upper nasal space without these additions and that could allow faster and more efficient programs to bring a wider range of drugs—and vaccines—to market.

## 1. Introduction

Deliberate nasal administration of drugs for medicinal purposes has a long history, was well documented in traditional Chinese medical practice [1], and is more extensively discussed in many Persian texts dating from the 9^th^ to 18^th^ centuries [2]. In recent history, it has been used primarily for treating local sino-nasal disease where systemic absorption was neither needed nor desired [3].

As more large molecule drugs were developed, especially the biological drugs in the last 10 years [4] and acute situations requiring self-administration and diseases with associated or comorbid gastrointestinal dysmotility were tackled, so non-invasive, non-oral delivery of drugs have become a more pressing need.

Nasal delivery started to gather more attention, however, as animal experiments demonstrated direct delivery of drugs to the brain [5,6,7,8,9], so-called “Nose-to-Brain” (or N2B) delivery, and it was speculated for humans [10], thus avoiding the difficulty that systemically administered drugs encounter of crossing the protective Blood Brain Barrier (BBB). This barrier effectively blocks high molecular weight (MW) substances, such as proteins and peptides [11] and even 98% of small molecular weight drugs [12] from access to the brain from the systemic circulation. There is emerging promising data bypassing the BBB with insulin, leptin, oxytocin and orexin A, but also hydrophilic molecules and charged molecules [13]. Indeed, even cells may be able to cross from the upper nasal space directly to the brain, as has already been shown in animals [11,14].

The data and opportunity were nicely summarized in a 2022 review [15], which reiterated many of the points made in a 2014 review [16], highlighting the slow progress, at least in clinical application, in the intervening 8 years. Indeed, one statement from the 2014 review, “evidence of direct N2B transport in man may still be considered lacking,” remains true, although differential delivery to the olfactory epithelium remains an important step in that direction which has now been achieved. The 2022 review nicely summarized the BBB and suggested how direct N2B delivery may be possible along branches of the trigeminal nerve, but most research indicates the olfactory nerve is the more promising access point. Both nerves may involve extracellular convective flow [17], transcellular [16], perivascular/paracellular [16,18], or perineuronal [16] routes. This 2022 review focused on the N2B delivery of insulin (and some other products) rather than the benefits of delivering to the olfactory epithelium of the UNS per se. Nanoparticles have been detected in the olfactory bulb 5 min after dosing [19], making N2B delivery potentially suitable even for acute indications. This interest in N2B has grown as more large MW drugs are developed for CNS indications, although low MW drugs can also benefit from rapid entry into the CNS. Parkinson’s disease (off episodes), epilepsy [20], migraine and acute agitation are all indications where drug penetration into the brain may be time-sensitive, and Alzheimer’s disease, multiple sclerosis and stroke have large MW candidates in development. For glioblastoma (direct access of cytotoxic drugs to the tumor may avoid undesired systemic toxicity. A comprehensively researched 2020 review [21] dealt with some of the device options being considered, specifically for N2B delivery, and those devices are summarized herein.

While interest in nasal delivery has gathered pace for high MW compounds, the COVID pandemic has shown that nasal mucosal immunity may not be sufficient with systemic vaccine administration to avoid mild disease or asymptomatic carriage and transmission. In addition to viral disease, the concept of being able to induce immunity to certain cancers with vaccination is also stimulating research [22]. There are now multiple nasal vaccine programs in development with the additional advantage that these vaccines may be more suitable for widespread use, where healthcare is less accessible or cold storage less available, and they may induce immunity at other distant mucosal surfaces [23]—e.g., the gastrointestinal and urogenital tracts.

Different approaches have been taken to target systemic disease by nasal drug delivery, with much interest and development focused on the addition of bioadhesives and absorption/permeation enhancers to formulations (especially liquid) delivered as a cloud to the lower nasal space. There are several excellent reviews of the range of excipients that have been investigated for these purposes [24,25], while others have focused on the drug-delivery systems more broadly [26,27,28]. An alternative approach has been to identify areas within the nose where absorption may be enhanced and deliver drugs to that area. This complimentary area of research has received much less attention, but with N2B delivery requiring non-invasive delivery to the olfactory epithelium of the upper nasal space, that interest is now growing and is the focus of this review.

## 2. Local Delivery

Local delivery within the nose has tended to deliver a cloud of liquid droplets (or powder particles) to the inside of the nostril. Much of the plume from most traditional options is actually deposited on the impermeable squamous epithelium of the nasal vestibule, with varying amounts penetrating to the mucus-covered respiratory epithelium of the lower nasal space.

### 2.1. Local Nasal and Sinus Disease

Many local sino-nasal diseases, specifically allergic rhinitis and rhinosinusitis with or without nasal polyps, have responded to the development of numerous topical corticosteroid sprays, starting with triamcinolone in 1957 but followed by beclomethasone, budesonide, flunisolide, fluticasone, mometasone and ciclesonide. All were delivered with either a pressurized metered-dose inhaler (pMDI) that was originally chlorofluorocarbon (CFC)-propelled but later changed to hydrofluoroalkane (HFA) or an aqueous product delivered by a “traditional” pump-spray.

Although many cases of chronic rhinosinusitis, especially with the additional pathology of nasal polyposis, are perennial, many patients have seasonal allergy problems making year-round administration of potent corticosteroids unnecessary. The target of the respiratory epithelium on the turbinates (conchae) of the LNS has been effectively reached by the traditional nasal spray delivering a diffuse cloud of droplets. In addition, systemic absorption is not desired, allowing for simpler formulations and delivery systems. Most recently, fluticasone has been approved for delivery by the OptiNose bidirectional Exhalation Delivery System (EDS) specifically for the treatment of chronic rhinosinusitis with nasal polyps (CRSwNP) [29] and is able to deliver more of the potent steroid “higher and deeper” into the nasal cavity (compared to traditional sprays) where many polyps originate [30]. The OptiNose device generally distributes better throughout the nasal cavity (than traditional nasal sprays) with its bidirectional design and is propelled by the patient’s own exhalation, which also elevates the vellum (soft palate), thus closing off the nose from the rest of the respiratory tract (Table 1). The nose piece directs the exhaled breath to push the drug product into one nostril, allowing it to reach the back of the nose, pass behind the septum and exit from the other nostril, hence creating the “bi-directional flow path” [16].

**Table 1 pharmaceutics-15-01720-t001:** Examples of Different Nasal Delivery Systems (in development or approved). The bottom four devices are basically syringes with different nozzles attached so that as a Healthcare Practitioner pushes the plunger of the syringe in, the liquid vaccine is forced out of the various tips creating a local mist or broad plume of particles 30–100 μm in diameter (in the case of the MAD).

Manufacturer	Device	Propellant	No. Doses	[Drug/Other Programs]	Pros	Cons		Reference/ Website
Aptar (Crystal Lake, IL, USA) (Device only) Multiple programs with drug companies	Unidose (UDS) Liquid Nasal Spray	None No spring. Finger squeeze pressure.	1 of up to 100 µL	Zomig^®^ [zolmitriptan] Zavzpret^®^ [zavegepant] Valtoco^®^ [diazepam] Tosymra^®^ [DHE] [+ several others + multiple partnered programs	One-handed operation. Low actuation force. No priming. No shaking. Multiple approved FDA/EMA products—for migraine (and other indications).	No distinction between UNS and LNS deposition. Diffuse “cloud” of droplets/particles.		[31]
	Bidose (BDS) Liquid Nasal Spray	None Spring	2 × 100 µL	Spravato^®^ [esketamine]				[32]
	Unidose (UDS) Powder Nasal Spray	None No spring. Finger squeeze pressure	1	Baqsimi^®^ [glucagon]	Ideal for low solubility molecules, minimizes excipient requirements, allows for larger dose administration, improves bioavailability and enhances drug diffusion/absorption. No shaking/priming.	No distinction between UNS and LNS deposition. Complex formulation and may increase irritability.		[33]
	Multi-Dose (preservative-free) VP3 Nasal Spray Systems	None	Multiple	Tyrvaya^®^ [varenicline]	The “traditional” nasal spray for >40 years. Mechanical system. Reference pump for originator anti-allergy brands and their generic alternatives.	No distinction between UNS and LNS deposition.		[34]
Bausch Health (Laval, Canada) (Drug device combination product)	Traditional Nasal Spray	None	1	Migranal^®^ [DHE]	Approved (>20 years) for delivery of 2.0 mg liquid DHE for acute treatment of migraines. Has a “traditional” pump spray.	Requires assembly and priming. No distinction between UNS and LNS deposition. Diffuse “cloud” of droplets/particles.		[35,36]
Haleon (Weybridge, UK)	Sensimist (liquid)	None	60 doses	Flonase^®^ [fluticasone] (generics and OTC focus)	Now OTC product. Designed to deliver a cloud of liquid fluticasone droplets to the LNS for allergic rhinitis. Systemic absorption is NOT desired.	Typical systemic corticosteroid effects are seen with long-term use and absorption.		[37]
Impel (Seattle, WA, USA) (Drug-device combination products)	Precision Olfactory Delivery (POD^®^) (Liquid)	HFA134a	1	Trudhesa^®^ [DHE] INP102 [insulin]	Focus on UNS for N2B.Device adapted for each product. Approved for delivery of 1.45 mg liquid DHE formulation for acute treatment of migraine	First Generation device—requires assembly and priming.		[38]
	POD (Powder)	HFA134a	Multiple exchangeable tips	INP103 [levodopa] INP107 [carbidopa/levodopa combination]	Several iterations of the device; research (which uses drug powder products in capsules) and single (no assembly) or multiple (changeable tip) dose design	Although delivering to UNS, there are no data on N2B delivery.Remains “in development.”		[39]
	POD (Powder)	Nitrogen	1	INP105 [olanzapine]	Single-use, preloaded device (no priming or assembly)	Research and development halted Q1 2023 (for business reasons)		[40]
Kurve (Lynnwood, WA, USA) (Device only)	ViaNase^TM^ (liquid)	None	1	N/A [insulin]	Focus on UNS for N2B. Device adapted for each product. Sixteen programs, fourteen in-clinic. Battery powered.	Unreliable performance in large NIH AD-insulin trial		[41,42]
OptiNose (Yardley, PA, USA) Drug-device combination products	Xhance^®^ (OptiMist^®^) Bi-directional	None (breath powered)	120 sprays	Xhance^®^ [fluticasone]	Approved for CRSwNP. Broader dispersion around the nasal cavity. The soft palate closes during use preventing the drug from entering the pharynx (or lungs).	Patient unfamiliarity with blowing into own nose. Initial priming is needed, and shaking before every dose. Focus on nasal disease—not systemic or N2B delivery.		[43]
	Bi-directional Exhalation Delivery System (EDS)—Onzetra^®^ Xsail^®^		1	Onzetra^®^Xsail^®^ [sumatriptan]	More of the drug reaches UNS. Approved for the delivery of sumatriptan powder (for migraine).	Patient unfamiliarity with blowing into own nose.		[44]
Satsuma (Durham, NC, USA) (Drug-device combination product)	STS101—a manually squeezed plastic bottle	None	1	N/A [DHE]	Small, portable, and no assembly or priming, delivering 6.0 mg DHE powder	Two failed phase 3 studies, with device modifications in between to improve delivery. No distinction between UNS and LNS delivery		[45]
AeroPump (Hochheim am Main, Germany) (Device only)	AeroPump	None		N/A [insulin] (adaptable for use by OTC and generic drugs)	Requires priming. Simple. Easy to use.	Non-targeted, LNS delivery		[46]
PharmaSystems (Markham, ON, Canada) (Device only)	Metered Nasal Dispenser	None	Multiple	N/A [insulin] (Canadian pharmacy supplier)	Requires priming. Simple. Easy to use. Programs showed benefits in postoperative delirium and postoperative cognitive function	Non-targeted, LNS delivery		[47]
MeadWestvaco (Richmond, VA, USA)/Silgan (Richmond, VA, USA) (Device only)	Mistette MkII Pump	None	Single/Multiple?	N/A [insulin] (provide bottles/pumps for liquid formulation delivery)	Requires priming. Simple. Easy to use.	Non-targeted, LNS delivery.		[48]
Nemera (La Verpillière, France) (Device only)	SP270+	None	Single/Multiple	N/A [insulin]	Requires priming. Simple. Easy to use.	Program was discontinued due to variations in viscosity.		[49]
SipNose (Yokneam, Israel) (Drug-device combination products)	Sipnose	Compressed air	Single and Multiple dose	N/A [insulin] (seven programs; five in-clinic, three of their own, and two partnered)	Performed well in Patient-Centered Care Impact Analysis [50] compared to two invasive ROAs (intrathecal and intracerebroventricular injection. Liquid and powder formulations. Targeted for UNS delivery.		Clinical data awaited	[51]
Teleflex (Wayne, PA, USA) (Device only)	Mucosal Atomisation Device (MAD)	None	1	N/A [vaccines] (Astra Zeneca: ChAdOx1nCoV-19 vaccine)	Fitted to the tip of a standard syringe in clinical trials of vaccines		Liquid (vaccines or drugs) administration by HCP	[52,53]
BD (Franklin Lakes, NJ, USA) (Device only)	Accuspray^TM^	None	1	N/A [vaccines] 510(k) cleared	Fitted to the tip of a standard syringe in clinical trials of vaccines		Liquid (vaccines or drugs) administration by HCP	[54,55]
Aptar	BiVax	None	1 (either 200 μL or 500 μL—split across 2 nostrils)	N/A [vaccines]	The liquid vaccine can be transferred directly from vial to applicator.		Liquid (vaccines or drugs) administration by HCP	[56,57]
Aptar	LuerVax (nozzle only)	None	1	N/A [vaccines]	Fitted to the tip of a standard syringe in clinical trials of vaccines.		For HCP liquid administration	

AD = Alzheimer’s disease; CRSwNP = chronic rhinosinusitis with nasal polyps; DHE = dihydroergotamine (for migraine); EMA = European Medicines Agency; FDA = Food and Drug Administration; HCP = healthcare practitioner; HFA = hydrofluoroalkane; LNS = lower nasal space; N/A = not applicable (no approved product); N2B = nose-to-brain delivery; NIH = National Institutes of Health; OTC = over the counter; POD = Precision Olfactory Delivery; UNS = upper nasal space.

### 2.2. Nasal Vaccination

The emergence in 2020 of COVID-19 as a global pandemic stretched public health systems globally but led to accelerated vaccine development programs. SARS-CoV-2, the novel coronavirus that causes COVID, as with all other airborne viruses, enters the body through the nose. Entry of the virus via the nose and, specifically, the olfactory mucosa of the nasal cavity is thought to be responsible for a number of the symptoms of COVID, including anosmia [58] and central nervous system effects [59]. Many vaccine development programs are now looking at nasal administration of vaccines to ensure mucosal, as well as humoral, immunity and appreciating that mass vaccination campaigns, especially in rural communities will benefit from vaccines that do not require refrigeration and/or administration by healthcare personnel using a needle. The rationale for nasal delivery of vaccines and the range of options now being considered is covered in Appendix A.

Nasal delivery of vaccines is not new, with three influenza (live attenuated) vaccines having received market approval in the US (Table 2) and a similar number in Europe, while another, Nasalflu (using an inactivated virosome), had to be withdrawn from the Swiss market [60]. However, the US Center for Disease Control reviewed data in 2016 and recommended the nasal Flumist vaccine (popular with children) not be used the following season due to poor results in the previous flu season, despite good results in previous years [61]. This highlights one of the issues with any vaccine for a seasonal disease, such as flu. Each year a new strain (or strains) emerge, usually in the Far East. Scientists must predict which will be the prevalent strain(s) in Europe/US the following season, and the vaccine must then be updated to deliver antigen from that strain or those strains. Sometimes those predictions turn out to be incorrect, and even the IM-administered vaccine can provide disappointing levels of protection.

A comprehensive summary of the nasal vaccines in development was recently published [62], citing nine ongoing programs against SARS-CoV-2, six against influenza, four against Respiratory Syncytial Virus (RSV), three against *Shigella sonnei*, two against *Bordetella pertussis* (whooping cough), with others against parainfluenza, norovirus, *Neisseria,* Ebola, anthrax, tuberculosis and HIV. In total, 36 current nasal vaccine development programs were registered, the vast majority, 31, were in phase 1 clinical development, three were in phase 2, one was in phase 4, and one was undisclosed. Details of the nasal delivery devices were minimal but suggested that 21 were simple nasal drops delivered by pipette or atomized from the tip of a syringe. A further eight used the term “nasal spray” but also appeared (according to the sponsoring organization’s website) to use a syringe with an atomizer tip (e.g., the Teleflex Mucosal Atomization Device, MAD Nasal^TM^ [52], BD Accuspray^TM^ [55] or Aptar’s LuerVax^TM^) (Table 1); one program compared the results obtained between vaccine delivered by “sprayer” and “pipette.” It was not possible to determine the delivery device in the remaining six programs, only that nasal delivery was planned for the vaccine. With intranasal delivery of vaccines already approved [63] and the knowledge that mucosal immunization drives CD8+ and CD4+ T cell responses [64] and leads to mucosal immunity, a better understanding of where in the nose vaccine payloads are delivered and then processed, is overdue. Aptar’s LuerVax delivered water droplets ranging in size from a Dv10 of 20 μm to Dv90 of 65 μm (meaning 10% of particles were smaller than 20 μm and 90% were smaller than 65 μm with a Dv50 of 35 μm [57] according to laser diffraction. This size range may be good for targeting the nasal turbinate, but for optimal delivery to the nasopharynx (where the Nasal Associated Lymphoid Tissue (NALT) is located, smaller droplets in the range of 7–17 μm are considered optimal [65]. For further information, see Appendix A.

## 3. Systemic Delivery

Most drugs, especially small molecules, are given orally, and this remains the most popular and widely used method of administration. When those drugs are peptides or proteins that will not survive transit through the gut; are largely metabolized on the first pass through the liver; are needed for situations where gastrointestinal absorption is delayed, such as in Parkinson’s disease [66] or migraine [67]; or if the drug is needed acutely, such as for seizure or anaphylaxis, then a non-oral route of administration is necessary—and preferably one that leads to rapid and reliable systemic drug concentrations. By avoiding oral administration, nasal delivery (to either the UNS or LNS) or pulmonary delivery with formulations of good bioavailability can often deliver microgram doses of a drug that access the systemic circulation, thus avoiding some of the adverse effects reported with milligram doses when administered orally. Intravenous injection is the gold standard, especially for acute situations, but when those situations occur in the community without immediate access to healthcare personnel, other options are required. Much focus has been on formulation changes that increase absorption (e.g., a powder desmopressin formulation improved on the liquid nasal formulation [68]) rather than where in the nose the drug is deposited. As such, many excipients and formulations have been tried; some with success, such as a dry-powder, chitosan-coated, liposome formulation loaded with ghrelin delivered with the Aptar UDS, which showed >50% deposition in the UNS using a 3D printed nasal cast model that was also better than the liquid formulation [69]. Further work has been conducted with thyrotrophin-releasing hormone [70], selective opioid agonist [19] and a chitosan-enhanced formulation of a modified peptide calcitonin gene-related (CGRP) antagonist [71]. Some of these programs look to explore N2B rather than systemic absorption and subsequent distribution to the brain.

### 3.1. Lower Nasal Space (LNS) Delivery

The delivery of drugs to the LNS can provide good bioavailability (BAV), i.e., access to the systemic circulation, with products such as Valtoco^®^ (diazepam) through the Aptar Unit Dose System (UDS) reported at 97% [72], although this good bioavailability was obtained with the help of dodecyl maltoside (DDM) as an absorption/permeation enhancer and Vitamin E to increase solubility. Another product with good nasal bioavailability is Nayzilam^®^ (midazolam), also delivered by Aptar’s UDS. Section 12.3 of the Prescribing Information states the absolute bioavailability to be approximately 44%. However, other intranasal midazolam formulations have been independently investigated [73], delivered by a different unit dose device obtained from Ing. Erich Pfeiffer GmbH, demonstrating better bioavailability ranging from 76 to 92%. These different formulations contained different concentrations of randomly methylated-β5-cyclodextrin (0, 2, 4, or 12%); different concentrations of saline and with or without 0.5% chitosan added as an absorption enhancer. In contrast, a very recent approval of Zavzpret^TM^ (vazegepant), also given by the Aptar UDS device [74], only reported ~5% bioavailability. The liquid formulation has the additional inactive ingredients: dextrose, hydrochloric acid, sodium hydroxide, succinic acid and water. In these examples, the same basic Aptar UDS delivery was used, presumably delivering to the same part of the nose (the LNS) with a diffuse nasal spray, showing that the addition of absorption enhancers was required to increase bioavailability. That, in turn, increases the complexity of the development program but does illustrate the extremes of the level of bioavailability that can be achieved through LNS delivery.

### 3.2. Upper Nasal Space (UNS) Delivery

There is now a greater appreciation that drugs delivered to the upper nasal space (UNS) may experience faster and more extensive absorption than when delivered to the LNS. Data were initially generated with the OptiNose^®^ bi-directional system and then, more recently, by several clinical programs using the Precision Olfactory Delivery (or POD^®^) system (Table 1), which specifically targets the upper nasal space (UNS). These programs have served as important steps as they highlighted the complex nasal architecture (Figure 1a) and showed one of the potential benefits of delivering drugs to the olfactory epithelium lining in the UNS. As can be appreciated, a cloud or broad plume of droplets delivered into the nose by a mechanical pump will coat the surfaces of the septum, inferior turbinate and lateral wall, with some penetrating to the middle turbinate but little to the superior turbinate and even less to the upper nasal space lying mostly above the superior turbinate. The nose is designed to convey volatile molecules to the olfactory epithelium in the UNS carried by inhaled air but not to allow a broad plume of droplets/particles to reach it. The olfactory epithelium has distinct differences from the respiratory epithelium of the nose’s humidification and filtering system covering the three turbinates (or conchae), mostly located in the lower nasal space (Figure 1b). These differences lead to very different absorption profiles for drugs landing on different mucosae [75]. Few data have historically been generated looking at where nasal devices deliver their payloads, with options being to investigate in cadavers, nasal cavity replicas, nasal casts or by using in vivo gamma camera imaging [76].

Investigating the novel breath-actuated, bi-directional system, Djupesland and colleagues compared the systemic levels of midazolam with those obtained after traditional nasal spray delivery and IV delivery [77]. The 100 μL of bespoke formulation, including cyclodextrin, HPMC, EDTA and benzalkonium chloride, to 12 healthy adult subjects showed similar serum concentration curves for the two nasal devices, both with rapid T_max_ of 15 min and a geometric mean ratio (OptiMist/Traditional spray) of 97.6%, suggesting no difference in systemic delivery between the two devices. However, this group went on to deliver ^99m^Technetium-labelled aerosol [78] to nine healthy subjects and showed increased delivery to the UNS, of 32%, versus 11% with a traditional spray. Subsequent work with the OptiNose^®^ system reported 53.6% of the formulation reaching the UNS (upper and middle posterior regions) versus only 15.7% with the traditional liquid spray [79] with no lung deposition.

Hoekman and colleagues generated data supporting the differential delivery systems to the UNS with Precision Olfactory Delivery (POD^®^) using MAG-3 (^99m^Technetium-labelled peptide) as determined by SPECT imaging in seven healthy subjects [80]. For this assessment, the nasal cavity was divided into four sections (Figure 1c): (1) vestibule, (2) lower nasal space, (3) UNS and (4) nasopharynx, determined from MRI imaging [81]. The UNS is where the olfactory epithelium is confined, which will be further discussed in the context of N2B delivery below—although there is also an approximately equal area of respiratory epithelium in the UNS, as it covers the superior turbinate. POD delivered substantially more (approaching 50%) to the UNS and less to the vestibule than the traditional spray (Figure 1d). The olfactory epithelium covers ~5 cm^2^, representing ~3% of the total surface area of the nasal cavity [82]. The enhanced delivery of radioisotopes to the UNS then led to several clinical programs being launched with the POD system. The STOP 101 [83] clinical study specifically looked at the delivery of the same liquid formulation through an approved, traditional spray to the lower nasal space and compared it with POD (using the same device that was subsequently used in the phase 3 study [84] and thereafter approved and commercialized) delivering it to the UNS and showed a four-fold increase in (C_max_), an almost four-fold increase in absolute bioavailability (59% vs. 15%) and an almost three-fold increase in AUC_0-inf_ (Figure 2a). To accomplish this, the POD system has a novel design for the nozzle (dosing tip), which focuses the plume of emitted liquid droplets into a narrow plume [85], which is pictured in the publication. The differences in plume and aerosol characteristics between the two systems, with the same formulation of dihydroergotamine (DHE) mesylate, were further explained in a technical manuscript [86] as well as the Anderson Cascade Impactor data and using Spraytec technology, with a particle size distribution of Dv10 ranging from ~290 μm, Dv50 ranging from 402–472 μm across 10 samples from two lots and Dv90 ranging from 561–734 μm—all well above the respirable range. In addition, the novel nozzle of the POD device allows patients to position the device in the correct orientation if they follow the Instructions for Use and insert the device up to the “shoulder” of the actuator arm, which was confirmed to be easy to achieve in human factor testing. 

To complete the clinical development for this INP104 program, the FDA required safety assessments of the UNS over 24 weeks with repeat dosing [84], which seemingly no other nasal delivery program had been asked to perform. This required the development of specific tools to assess the mucosal integrity of the olfactory epithelium and olfactory function, which detected no significant adverse effects over 24 and even 52 weeks [87].

A subsequent program, INP105, with a spray-dried powder formulation of olanzapine [88], showed in phase 1 a similar C_max_ and AUC as the same dose administered by intramuscular injection but with a faster T_max_; however with no approved IV formulation of olanzapine to compare against, absolute bioavailability could not be determined in this study (Figure 2b). The POD powder programs (both INP105 and INP103/107 with levodopa/carbidopa) [89] benefited from the previous development of both rodent and primate versions of the POD which expedited formulation development and pharmacokinetic characterization before, or even during, human trials [90], with the structure and function of the non-human primate nose being most similar to that of humans [91].

**Figure 1 pharmaceutics-15-01720-f001:**
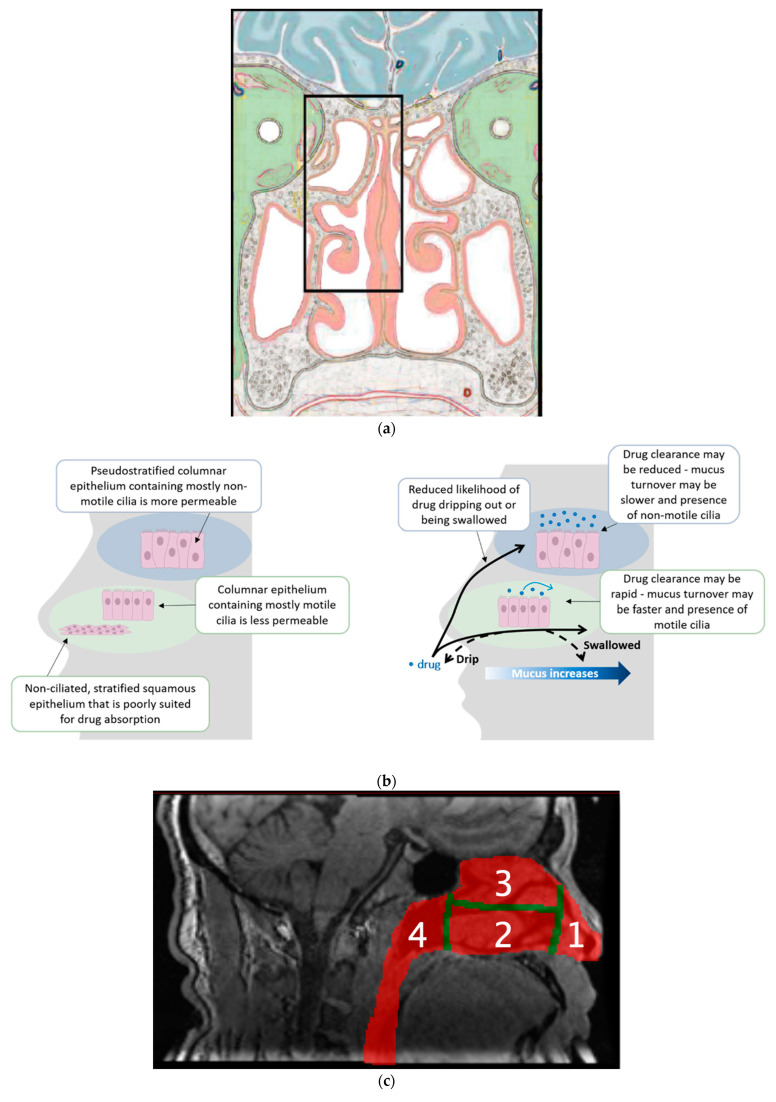
(**a**) Cross-section of the frontal portion of the human head, representing the posterior third of the nasal cavity, with outlined region (black box) highlighting areas commonly enriched with olfactory epithelium. Adapted from Salazar, I.; Sanchez-Quinteiro, P.; Barrios, A.W.; López Amado, M.; Vega, J.A. *Handb. Clin. Neurol.* **2019**, *164*, 47–65, with original adaptation from Schünke, M.; Schulte, E.; Schumacher, U. et al. *Prometheus: Texto y Atlas de Anatomía.* 3ª edición, vol. 3. Madrid: Panamericana; 2014 [19]. (**b**) Diagram showing the different epithelium (left panel) and clearance mechanisms (right panel) of the nose [75]. Reproduced with permission from the publisher. (**c**) SPECT Imaging data of nasal delivery of MAG-3 (^99m^Technetium-labeled peptide) by POD vs. a traditional nasal spray in 7 Healthy Volunteers. To determine nasal deposition, the nasal cavity was divided into four regions: (1) Vestibule, (2) Lower nasal space, (3) UNS and (4) Nasopharynx based on MRI imaging [58]. (**d**) POD delivery led to significantly (* *p* < 0.05) greater deposition in the UNS (and nasopharynx) and less deposition in the vestibule (than the traditional spray) [81]. Regions defined in (**c**).

**Figure 2 pharmaceutics-15-01720-f002:**
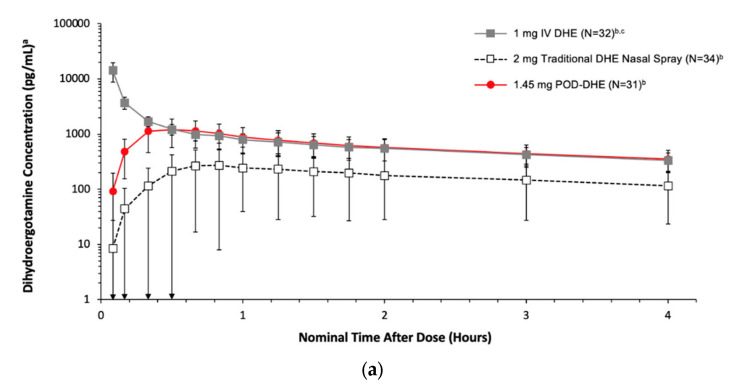
(**a**) STOP 101: Mean plasma DHE concentrations 0 to 4 Hours postdose (Safety Population) [61]. ^a^ Measures dihydroergotamine free base on a semi-log scale; ^b^ Doses represent dihydroergotamine mesylate; ^c^ N = 31 for time points of 5, 10, 30, and 40 min. Note: Dihydroergotamine concentration measurements begin at the 5-min timepoint. Represented as mean (SD). For the calculation of mean values, individual BLQ values were set to zero before determining the mean. BLQ = below the limit of quantitation data; IV = intravenous; SD = standard deviation. (**b**) SNAP 101: Plasma olanzapine concentrations 0 to 2 h post-dose [86]. INP105 = POD OLZ.

## 4. Nose-to-Brain (N2B) Delivery

In the last decade, some big companies exited drug development for neurological indications [92], but research and development in CNS indications has seemingly now picked up again [93] as CNS diseases are increasingly recognized as the leading cause of disability globally [94]. The challenge remains that at least two approved new therapies, both antisense oligonucleotides developed by Ionis Pharmaceuticals for devastating neurological diseases—spinal muscular atrophy [95] and amyotrophic lateral sclerosis [96], have to be delivered by intrathecal injection. So, the search continues for a less invasive way to circumvent or breach the Blood Brain Barrier (BBB). A comprehensive review of the direct N2B delivery of therapeutic peptides was published in 2022 [97]. To summarize that work: The therapeutic peptide market is large (US$39.3 Billion in 2021) but is expanding rapidly and is due to reach nearly $91.25 Billion in 2031 [98], while the nasal delivery market is also rapidly expanding (from $7.8 Billion in 2018 to $12 Billion by 2026 [99]. The article listed 14 approved small molecule products delivered nasally and 21 large molecular weight (MW) products (of which 9 were various versions of desmopressin), including one attenuated live virus vaccine (Flumist^®^). Of those large MW products, only three have received approval in the last 10 years, and only 1 of those, glucagon (Baqsimi^TM^) in the last 5 years. This slow pace of translating the exciting preclinical data into clinical programs is disappointing, but at last, that situation may be changing with ongoing research programs in several indications where interesting data has been generated: insulin in cognitive impairment/Alzheimer’s disease [100] in 2017; oxytocin to improve social communication in autism spectrum disorders [101] 2018; orexin-A in narcolepsy [102] 2011; basic fibroblast growth factor in Parkinsonism [103] 2021; leptin in obesity [104] in 2018, and there remains vigorous activity in oligonucleotide development for neurological disease [95,96]. However, clinical progress still remains slow, with direct injection of growth factor protein into the brain proposed as remaining the most promising option [103], with the development of protective biomaterials and device selection still required to better deliver these peptides across the BBB in a less invasive manner.

In his 2014 paper [16], Djupesland reviewed many of the articles claiming N2B transport, commenting then that encouraging data had been generated with both insulin and orexin, based on MRI imaging and/or clinical response, but that “convincing, unequivocal proof of N2B transport was still missing in humans, largely due to methodological and ethical considerations”. Sadly, 9 years later, we are really no further forward.

Insulin N2B has been the most studied large molecular weight (5808 Da) peptide [100,105,106,107,108,109,110], perhaps not surprising since the discovery of specific insulin receptors in the olfactory bulb, hippocampus, hypothalamus and lower brainstem [111].

A 2018 study [109] looked at data from 38 clinical studies of acute nasal insulin administration that had dosed 1092 individuals and a further 18 studies in 832 individuals for periods between 21 days and 9.7 years and found no evidence of symptomatic hypoglycemia or serious adverse events, while 10 of the acute studies reported minor, temporary reductions in blood glucose and one report of minor glucose reduction in the chronic study. Overall blood glucose fell by 0.2–0.5 mmol/L (4–9 mg/dL) and lasted longer than the temporary rise in circulating insulin [110]

Another review from 2021 similarly looked at the growing number of clinical trials investigating N2B delivery [111]. One such device claiming UNS delivery and showing promising results with the delivery of insulin is the Kurve ViaNase^®^ device leading to the Study of Nasal Insulin to Fight Forgetfulness (SNIFF) study, a large phase 3 NIH trial [42]. Kurve’s ViaNase technology incorporates an electronic atomizer that creates a vortex of nebulized particles, their Controlled Particle Dispersion (CPD^®^) nasal delivery technology to “create a precisely-controlled turbulent flow” appropriate for monoclonal antibodies and larger peptides, maximize distribution to the UNS and minimize pharyngeal deposition. Several publications have featured Kurve’s technologies, with one of those publications [112] reporting, in 2015, an exploratory study delivering a single dose of 2-PMPA (a blocker of the brain metallopeptidase, glutamate carboxypeptidase II, thought to be responsible for excess glutamate in neurodegenerative diseases) to a single non-human primate. At 30 min post-dosing, 2-PMPA was undetectable in the plasma (lower limit of quantitation being 50 nM), while the concentration in the CSF was 0.32 μg/mL (~1.5 μM) determined by LC/MS/MS. Several ^99^Technetium labeled scintigraphic images on their website (kurvetx.com/research/peer-review-deposition comparison, dated 2005) show the widespread distribution of isotope throughout the nasal cavity, but it is not clear that this data was, in fact, peer-reviewed or if it has ever been published.

Kurve technology was selected for the NIH SNIFF trial, but the device suffered from frequent malfunctions and after 49 patients were dosed, it was replaced with the POD device for the remaining 240 patients with mild cognitive impairment [42]. The trial failed to show clinical benefit compared to placebo, but there were no reports of hypoglycemia, suggesting a lack of significant systemic absorption of the insulin. Kurve’s ViaNase technology is still being advanced for treating psychiatric conditions, post-stroke and cognitive impairment in multiple sclerosis, with six separate programs delivering insulin in mild cognitive impairment, five programs delivering polyclonal antibodies in neurodegenerative disease, and five other programs featured on their corporate website. Other N2B delivery projects have looked at cholecystokinin [113], erythropoietin [114], melanocortin [115], glutathione [116], perillyl alcohol [117], angiotensin II [118] and neurotrophic factors [119] all with promising preclinical, or clinical case study results but have yet to be subjected to large scale, randomized clinical trial investigation.

While the importance of delivery of these drugs and peptides to the olfactory epithelium for N2B was stressed, there is little data comparing UNS delivery to LNS delivery to demonstrate this. Neither is there clarity about what device was actually used to generate the data. This highlights one of the important questions still to be addressed in clinical development: how to demonstrate N2B delivery in humans before investing in and conducting long-term clinical studies.

### 4.1. Olfactory Delivery

Millions of Olfactory Sensory Neurons (OSNs) are embedded in the olfactory epithelium and have long, non-motile cilia extending into the UNS. This is the only place in the body where the CNS is in direct contact with the environment [16,120]. These OSNs die and regenerate—perhaps the best-known neurons with that ability likely resulting from their exposure to inhaled chemicals—with a lifecycle of 30–60 days [15], although much of the research showing this comes from mice [121]. Once drugs are delivered onto the UNS, they may be transported across the olfactory epithelium inside the nerve axon (intracellularly), between the epithelial cells (paracellularly) or through the epithelial cells (transcellularly) [16], and thereafter the drug enters the olfactory bulb and then may distribute to other areas of the brain. Uptake by the OSN is by either non-specific or receptor-mediated endocytosis, predominantly the former [111]. The endocytosed drug is then transported to the olfactory bulb, taking an estimated 0.74–2.67 h [15,122]. Data from mice has shown drug in the CNS by 5 min post administration [123], peaks in the olfactory bulb by ~10 min [15] and distributes to distant brain regions such as the hypothalamus and midbrain by 30 min [124]. As the OSNs apoptose, so there are temporary gaps between the sustentacular cells of the epithelium, through which drug can pass—the paracellular route—and enter the perineural space that surround the OSNs as they traverse the cribriform plate [125]. Once in the brain, distribution may be by continued intracellular transport, although considered more likely is ongoing extracellular transport by convective bulk flow and a putative “perivascular pump” [15]. It is important to remember that many of the experimental studies conducted in rodents (whose olfactory epithelium may cover 50% of their nasal cavity [126] rather than the ~5% in humans) will be conducted by highly trained and familiar technicians applying drugs to anesthetized animals lying on their backs. The olfactory epithelial surface area to body weight ratio in humans is very different from those found in other animal species [127] (Table 3), and unanesthetized humans will be physiologically different from anesthetized animals. These considerations make it challenging to extrapolate animal data to humans, and the usual allometric scaling conventions applied when moving from preclinical to clinical experiments may not apply. Hence knowing both the surface area of an animal’s olfactory mucosa and its surface area to weight ratio may need to be considered prior to clinical studies. As yet, there is no consensus on how to measure the direct CNS uptake of drugs in humans, short of the longer-term efficacy benefits that all drugs seek. What is known, however, is that cerebrospinal fluid levels obtained by lumbar puncture may not accurately reflect tissue levels in the brain [12,128]. As the route of administration of intrathecal drugs, such as the oligonucleotides Spinraza^®^ [95] and Qalsody^®^ [96], is also through a lumbar puncture, there may be other excellent therapeutics that, even with intrathecal delivery, show disappointing clinical results due to the failure to get from the CSF into the CNS tissue due to the protective arachnoid membrane and its tight junctions [129] with only ~5% of the CSF reaching and circulating around the olfactory region [12]. It is estimated that the dose delivered by this route may only need to be 0.01–1% of an oral dose [125].

### 4.2. Trigeminal Delivery

The trigeminal nerve provides sensory innervation to the LNS through its ophthalmic (V1) and maxillary (V2) branches. These supply upper anterior segments of the nasal space (i.e., also supply areas that the olfactory nerve serves, through its OSNs) and lower parts of the nasal space, respectively, and provide the respiratory epithelium with both sensory and parasympathetic innervation, projecting back to the pons. Unlike the OSN, the trigeminal nerve endings do not project above the epithelial surface; thus, any trigeminal transport first has to cross this respiratory epithelium or pass between the tight junctions of the mucosal cells to reach the underlying neurons. This results in transport via the trigeminal nerve being slower than via the olfactory nerve, estimated at 17–56 h [15] from mucosa to brain, with the axonal transport component alone also being slower (than olfactory transport) at 3.7–13.3 h [120]. However, even some drugs delivered to the LNS may be taken up by these neurons and transported directly to the brain. That could be the reason why one traditional nasal spray administered vaccine containing a neurotoxin resulted in dozens of cases of Bell’s palsy and had to be withdrawn from the (Swiss) market [60]. Whether this route will be able to supplement or supplant olfactory delivery remains unknown at this time, and similarly unknown is whether future imaging technology will be able to differentiate between the two neuronal routes.

### 4.3. Vomeronasal Terminalis

The vomeronasal organ (VNO) [130] is present in newborn infants in the soft tissue at the base of the nasal septum just above the hard palate and is the termination of the vomeronasal terminalis nerve [79]. The VNO (or Jacobson’s organ) is fully functional in the macrosmotic rodents and canines, sensing not only compounds emanating from prey and predators but sex pheromones [131] too, but is vestigial in human adults. It is unknown how much, if any, drug may reach the olfactory bulb via this nerve.

## 5. Discussion

As medicine advances, and especially as many CNS diseases are now receiving much greater attention, so focus on how to deliver these new medicines has increased. Drug delivery has to address the inability of these medicines to be delivered orally and how to cross the BBB. Only ~2% of small molecule drugs cross the BBB from the bloodstream, and maybe only 0.1% reach the CNS [132], so it is only suitable for potent molecules, especially peptides and proteins. With advances in bioadhesives, absorption/permeation enhancers and other Drug-Delivery Solutions (such as the addition of alkylsaccharides [133]—as used in Valtoco—see Table 1) and new devices, the non-invasive delivery of more of these future medicines has become possible.

Emerging infectious diseases, such as COVID that enter the body through the nose—can still cause illness and be transmitted in subjects who have received systemic vaccination, making nasal vaccine delivery and mucosal immunity a higher priority [134]. So far, vaccine programs have not further investigated where in the nose their payload is delivered or whether delivering to different parts of the nose affects the immunological response. If a vaccine were to be delivered to the olfactory epithelium, depending on its components, it too could gain direct access to the CNS, which could be detrimental, although allowing a vaccine to dwell longer on the epithelium could aid uptake into the Nasal Associated Lymphoid Tissue (NALT) and thus confer greater mucosal protection for the whole nasal mucosa. To date, most vaccine programs have utilized traditional nasal aerosol sprays or “nose drops,” which deposit most of their contents in the non-absorptive (squamous epithelium lined) vestibule or respiratory epithelium lined lower nasal space [135].

The UNS is also the target for drugs seeking N2B delivery, and while imaging (MRI, PET, SPECT and gamma scintigraphy [21] can provide some supportive data and all show promise, none have yet to be fully validated as a way to measure deposition into the UNS, let alone absorption into the CNS although PET/MRI is considered most sensitive at quantifying N2B delivery [135]. Current options to assay the drug levels remain HPLC, high sensitivity mass spectrometry and immunoassay [127], while other methods to determine direct N2B delivery are also being considered, such as changes in brain metabolism [108], selective insulin impairment [136], changes in brain blood flow [137] and neuromodulation [138]. These options need to be vigorously explored so that the requisite evidence of direct N2B delivery in humans can be convincingly demonstrated.

Previous nasal delivery systems have not proven to be very popular for a variety of cultural or clinical reasons: unfamiliarity, local intolerability (e.g., drug dripping down the back of the throat, bad taste and swallowing), drug loss as it drips out of the front of the nose (requiring lying head hanging to combat gravity), or lack of consistency of absorption and thus effect. Patient adherence is critical, and thus formulation for taste should be considered early in development with patient input to the design of devices, their instructions for use and important Human Factor work undertaken prior to commercialization. Different strategies have been employed to try and overcome these difficulties, with varying degrees of success, but only recently has attention shifted to delivering drugs to different parts of the nose, specifically to land on the olfactory epithelium of the UNS for drugs requiring rapid and extensive absorption, or potentially N2B.

Targeting the olfactory epithelium may, by virtue of its non-motile cilia and high vascularity, also provide potentially highly differentiated systemic PK profiles for drugs compared to traditional nasal delivery and may be better than intramuscular injection and only surpassed in efficiency by intravenous injection, at least for some, maybe simpler nasal formulations. Conditions where acute treatment is required, such as seizures, migraine, agitation, off episodes in Parkinson’s disease and anaphylaxis, could well benefit from drugs redirected to this epithelium when access to a healthcare practitioner and the requisite equipment for IV injection or infusion are not available. Delivery to the UNS may be the optimal target within the nose for small- and large-molecule drugs and perhaps for some vaccines. Further information about mucosal immunity and vaccine adjuvant and excipients, nanotechnology and the surge of nasal vaccine delivery clinical trials [139,140,141,142,143,144,145,146,147,148,149,150,151,152,153,154,155,156,157,158,159] can be found in the Appendix A.

## 6. Conclusions

Nasal delivery of drugs and vaccines using small, portable, easily operated (by patients themselves, caregivers or healthcare professionals) devices will likely never replace oral pills, capsules and tablets. But where rapid drug levels are required, oral drugs are prone to gut or liver degradation, disease-related gastrointestinal dysfunction or comorbidity delay absorption, large molecules need to be delivered, delivery of drugs N2B for CNS disease is being contemplated, or induction of nasal (or distant) mucosal immunity by vaccinations is required, there are compelling reasons to consider nasal delivery and as we understand more about the complex anatomy and physiology of this organ, perhaps targeting the UNS within the nose. I believe that in the years to come, nasal delivery will become much more popular for all three situations: non-invasive systemic delivery, N2B delivery, and ensuring broad mucosal immunity. Regional deposition within the nose will thus become a standard part of development programs, along with an array of formulations to ensure safe, effective and consistent delivery of their large molecule and even future cellular cargo.

## Figures and Tables

**Table 2 pharmaceutics-15-01720-t002:** Currently US-approved nasally administered vaccines (US).

Vaccine	Approved	Manufacturer	Target	Antigen	Adjuvant & Excipients
Influenza A (2009) Monovalent	2009	MedImmune (Gaithersburg, MD, USA)	H1N1 Influenza	Cold-adapted, live attenuated virus: A/California/7/2009 H1N1v	Monosodium glutamate; porcine gelatin; arginine; sucrose; potassium phosphate; gentamicin.
FluMist^®^ (Trivalent)	2003	MedImmune	3 Influenza strains	Cold adapted, live attenuated virus: A/California/7/2009(H1N1); A/Perth/16/2009(H3N2); B/Brisbane/60/2008	Monosodium glutamate; porcine gelatin; arginine; sucrose; potassium phosphate; gentamicin.
FluMist^®^ Quadrivalent	2003	MedImmune	2 Influenza A strains and 2 B strains	Cold adapted, live attenuated virus: A/Victoria/1/2020(H1N1); A/Norway/16606/2021(H3N2); B/Phuket/3073/2013; B/Austria/13599417/2021	Monosodium glutamate; porcine gelatin; arginine; sucrose; potassium phosphate; ovalbumin; gentamicin; ethylenediaminetetraacetic acid (EDTA).

**Table 3 pharmaceutics-15-01720-t003:** Olfactory epithelium surface area:Body weight ratio (cm^2^/kg) of different species. Adapted from [130].

Species	Olfactory Epithelium Surface Area (cm^2^)	Olfactory Epithelium Surface Area:Body Weight Ratio (cm^2^/kg)
Mouse	1.25–1.40	31.0–35.0
Rat	4.20–6.80	12.0–19.0
Dog	170.0–380.0	17.0–38.0
Small primate	0.25–0.55	2.30–2.75
Human	10.0	0.14

## Data Availability

Not applicable.

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
