# Peer review of "The Upper Nasal Space: Option for Systemic Drug Delivery, Mucosal Vaccines and “Nose-to-Brain”"

_pharmaceutics, 2023, doi:10.3390/pharmaceutics15061720_

Round 1

Reviewer 1 Report (Previous Reviewer 3)

Thanks for allowing me to review this manuscript. I feel that the authors have addressed the reviewer’s comments in detail. Overall, the manuscript has been much improved compared to the earlier version.

My only question is on page 15: line 287: "showed a four-fold increase in bioavailability (Cmax)"- is it four fold increase in peak concentration (Cmax) or the bioavailability?    

Author Response

Clarified: 4-fold increase in Cmax AND almost 4-fold increase in absolute bioavailability (59% vs 15%) and almost three-fold increase in AUC0-inf.

Reviewer 2 Report (New Reviewer)

The authors revised the manuscript for clarity.

Author Response

Thank you.

Reviewer 3 Report (New Reviewer)

The manuscript contains novel and well collected information about nasal delivery and medical devices, although it is difficult to follow, because of numerous modifications indicated by track changes, its content was significantly changed in comparison to possible previous version. Both local and systemic utilization of nasal delivery route is detailed described according to different anatomic part of nasal cavity, however some additional parameters should be taken into consideration to discuss:

How the plume geometry and spray cone angle effects on the deposition and how it can be exploited for controlled nasal transport?

Also, the administration angle represents an important parameter in deposition, which requires patient education and high degree of adherence.

Author Response

How the plume geometry and spray cone angle effects on the deposition and how it can be exploited for controlled nasal transport? I agree this is important information. However the plumes (of DHE) when delivered by POD and traditional spray have already been published (text and reference added, Silberstein et al) and a more technical characterization of the plume also published (text and reference added, Cooper et al)

Also, the administration angle represents an important parameter in deposition, which requires patient education and high degree of adherence. Agreed – but arguably less due to the biphasic nature of the POD delivery system with the following stream of propellant pushing the delivered droplet (or particles) deeper into the upper nasal space. Text added.

Reviewer 4 Report (New Reviewer)

Manuscript ID: pharmaceutics-2414525

Type: Review

Title: The Upper Nasal Space – an Option for Systemic Drug Delivery, Mucosal Vaccines and “Nose-to-Brain”?

Author: Stephen Bevan Shrewsbury*

Stephen B. Shrewsbury reviewed a topic entitled “The Upper Nasal Space-an Option for Systemic Drug Delivery, Mucosal Vaccines and “Nose-to-Brain”?” The author focused on the recent progress about the intranasal route of administration that is gaining increasingly relevance as non-invasive approach for the treatment of neurological diseases and, in particular, the neurodegenerative ones.

The study included an introduction where it is stated the focus of the Review and namely to identify areas within the nasal cavity where absorption may be enhanced and deliver drugs to these areas. Next, in a second part, the study deals with the issue to treat local nasal diseases or to obtain nasal vaccination and it can be accomplished targeting the respiratory epithelium of the Lower Nasal Space (LNS). In the third part, the author faces the issue to achieve systemic delivery and it is possible by targeting both LNS and Upper Nasal Space (UNS). Finally, the author deals with Nose-to-brain delivery devices which allow to maximize distribution to the UNS. The main conclusion of this review is that nasal administration of drugs and vaccines will likely never replace oral delivery systems as tablets and capsules. However, when rapid drug levels are required or oral drugs can undergo extensive degradation in gut and liver, the nasal delivery should be considered. In my opinion, the novelty of this review lies in the important information provided on the delivery technologies (in development or approved) to be employed for targeting specific areas within the nasal cavity, a neglected aspect from other reviews in the field.

Overall, the article is well-organized and without any confounding digressions. However, sometimes it is excessively concise to the point that important information is missing giving the impression to be a “mini-review”. This is a weakness of the review together with the high number of references to papers with more than five years old occurring in it. I recommend a revision addressing the following major and minor issues carefully before publication also in order to reach a broader audience.

1.                  It should be interesting to know how the author justifies the high number of references to papers with more than five years old occurring in his review. The author could also consider if it is appropriate to report such justification in the text.

2.                  Section 3.2. Upper Nasal Space (UNS) delivery. In my opinion, in this section a more detailed discussion about the important delivery system named POD (allowing targeting of UNS) is missing. It is true that a representation of POD is reported in Table 2, but I believe it not inappropriate to report in the text some technological details about this delivery system. What is the author’s opinion about the reason for which POD device allows to reach the Upper Nasal Space (UNS)? Please, explain in the text.

3.                  In connection with that reported in the previous point, I don’t see inappropriate to spend a little space of the text to technological details of other important devices such as Aptar’s UDS, ViaNase®; OptiNose® and Mucosal Atomization Device (MAD), if possible of course.

4.                  Line 247 Figure 1a. In my opinion, the author should better clarify and report in the text a more detailed discussion of this Figure and why it highlighted the complex nasal architecture. Please, clarify this aspect. In the present form it is unclear.

5.                  Figure 2b. In this Figure it is reported OLZ ODT. I know that ODT is the acronym of “oral disintegrating tablet” but it seems to me that the author has not defined it previously in the text. Please check this aspect.

6.                   Line 561 Table 3 Olfactory epithelium surface area: In this Table are reported: Olfactory epithelium surface area (cm2) and Olfactory epithelium surface are(a): Body weight ratio (cm2/kg) of different animal Species. While I fully understand the reason to include the Olfactory epithelium surface area (cm2), I don’t rationalize the insertion of the ratio Olfactory epithelium surface are(a)/Body weight ratio (cm2/kg). What is the significance of this ratio? Why is it important to evaluate this ratio? Please, explain in the text. Please, consider that a normal reader of the journal may have not access to the reference cited by the author for Table 3 (i.e., Ref [97]).

7.                  L. 588 Summary. I don’t understand why it is reported at this point a paragraph entitled “Summary”. Please, clarify whether this paragraph corresponds to the “Abstract” of the review (and, if so, it should be located before the “Introduction”) or not.

8.                  L. 852 Conclusion. I agree with the author’s main conclusion i.e. when rapid drug levels are required or oral drugs can undergo extensive degradation in gut and liver, the nasal delivery should be considered. However, I believe that the potential of N2B approach is still to be fully clarified, particularly for the neurodegenerative diseases. I still don’t rule out that N2B may have a relevant role in these cases being a non-invasive method to effectively by-pass the BBB. In the conclusive paragraph, it may be appropriate to know the personal author’s opinion about N2B even in perspective view.

9.                  L. 852 Conclusion. In this paragraph there is no comment of the author about the question reported in the title and namely “The Upper Nasal Space – an Option for Systemic Drug Delivery, Mucosal Vaccines and “Nose-to-Brain”?” In my opinion, the author should report in the text his opinion on this question even in perspective view. I remember herein that according to “Instructions for Authors” of the Journal, Review manuscripts should ”offer a comprehensive analysis of the existing literature within a field of study, identifying current gaps or problems. They should be critical and constructive and provide recommendations for future research.”

Author Response

  1. It should be interesting to know how the author justifies the high number of references to papers with more than five years old occurring in his review. The author could also consider if it is appropriate to report such justification in the text. See additional text to explain this in Section 4. I agree that it is a shame the exciting preclinical data that was (as the reviewer notes) published more than 10 years ago, has NOT translated into more clinical programs featuring nasal delivery – and indeed to more clarity about what is needed to demonstrate N2B in humans. I am sure that will change with the number of nasal delivery programs at last increasing.
  2. Section 3.2. Upper Nasal Space (UNS) delivery. In my opinion, in this section a more detailed discussion about the important delivery system named POD (allowing targeting of UNS) is missing. It is true that a representation of POD is reported in Table 2, but I believe it not inappropriate to report in the text some technological details about this delivery system. What is the author’s opinion about the reason for which POD device allows to reach the Upper Nasal Space (UNS)? Please, explain in the text. I have attempted to summarize some of the aerosol and plume data – which has been published elsewhere and those references (Silberstein & Cooper added)
  3. In connection with that reported in the previous point, I don’t see inappropriate to spend a little space of the text to technological details of other important devices such as Aptar’s UDS (does not specifically claim UNS delivery but have reached back out to Aptar again for more information on plume, particle size, plume etc ), ViaNase® (detail added – section 4 from company website – and publications referenced ; OptiNose® (additional detail from Djupesland’s publication added) and Mucosal Atomization Device (MAD), (more detail added in text and in Table 2) if possible of course.
  4. Line 247 Figure 1a. In my opinion, the author should better clarify and report in the text a more detailed discussion of this Figure and why it highlighted the complex nasal architecture. Please, clarify this aspect. In the present form it is unclear. (more detail added in Section 3.2)
  5. Figure 2b. In this Figure it is reported OLZ ODT. I know that ODT is the acronym of “oral disintegrating tablet” but it seems to me that the author has not defined it previously in the text. Please check this aspect. Corrected – thank you.
  6. Line 561 Table 3 Olfactory epithelium surface area: In this Table are reported: Olfactory epithelium surface area (cm2) and Olfactory epithelium surface are(a): Body weight ratio (cm2/kg) of different animal Species. While I fully understand the reason to include the Olfactory epithelium surface area (cm2), I don’t rationalize the insertion of the ratio Olfactory epithelium surface are(a)/Body weight ratio (cm2/kg). What is the significance of this ratio? Why is it important to evaluate this ratio? Please, explain in the text. Please, consider that a normal reader of the journal may have not access to the reference cited by the author for Table 3 (i.e., Ref [97]). Edited to address: These considerations make it challenging to extrapolate animal data to humans and usual allometric scaling conventions applied when moving from preclinical to clinical experiments may not apply. Hence knowing both the surface area of an animal’s olfactory mucosa, and their surface area to weight ratio, may need to be considered prior to clinical studies
  7. L. 588 Summary. I don’t understand why it is reported at this point a paragraph entitled “Summary”. Please, clarify whether this paragraph corresponds to the “Abstract” of the review (and, if so, it should be located before the “Introduction”) or not. Edited title of this section to Discussion and edited the content
  8. L. 852 Conclusion. I agree with the author’s main conclusion i.e. when rapid drug levels are required or oral drugs can undergo extensive degradation in gut and liver, the nasal delivery should be considered. However, I believe that the potential of N2B approach is still to be fully clarified, particularly for the neurodegenerative diseases. I still don’t rule out that N2B may have a relevant role in these cases being a non-invasive method to effectively by-pass the BBB. In the conclusive paragraph, it may be appropriate to know the personal author’s opinion about N2B even in perspective view. Final sentence to conclusion added.
  9. L. 852 Conclusion. In this paragraph there is no comment of the author about the question reported in the title and namely “The Upper Nasal Space – an Option for Systemic Drug Delivery, Mucosal Vaccines and “Nose-to-Brain”?” In my opinion, the author should report in the text his opinion on this question even in perspective view. I remember herein that according to “Instructions for Authors” of the Journal, Review manuscripts should ”offer a comprehensive analysis the existing literature within a field of study, (more references reviewed and added) of identifying current gaps or problems (demonstrating, convincingly, N2B in humans remains elusive – but I have discussed the options in para 2 of the discussion). They should be critical and constructive and provide recommendations for future research.” Final sentence to conclusion added: I believe that in the years to come nasal delivery will become much more popular for all 3 situations: non-invasive systemic delivery; N2B and broad mucosal immunity.

Reviewer 5 Report (New Reviewer)

The review summarizes the feasibility of the Upper Nasal Space – an Option for Systemic Drug Delivery, Mucosal Vaccines and “Nose-to-Brain”. Although the review contains many valuable data. However, it needs many modifications before being accepted

Major comments

1. The review contains many vague expressions that are difficult to be understood specially in the first paragraph of the introduction

2. Table 2 should be revised, many data are missed and overlapping.

3. Summary Section should be revised not to repeat your previously mentioned description in previous section. I suggest replacing this section by future directions and Expert opinion Section.

Minor comments

Why adding a question mark in the tile?

Line 26-27, rephrasing is required

Line 31-32, check foe meaning

33-36, revise your description

296, correct BloodBrainBarier

Line 300, correct $12 Billion by 2026[73[NasalDrug].

Line 415, correct CNS[102Mistry]

Line 439, correct PET, SPECT and gamma scintigraphy [21Trevoni] can…

Author Response

The review summarizes the feasibility of the Upper Nasal Space – an Option for Systemic Drug Delivery, Mucosal Vaccines and “Nose-to-Brain”. Although the review contains many valuable data. However, it needs many modifications before being accepted

Major comments

  1. The review contains many vague expressions that are difficult to be understood specially in the first paragraph of the introduction. I have tried to restructure the introduction but am finding it hard to identify the “vague expressions” that the reviewer notes.
  2. Table 2 should be revised, many data are missed and overlapping. As stated the table is not meant to cover ALL nasal delivery technology, but to provide EXAMPLES and to focus on those that deliver to the UNS – OptiNose, Kurve ViaNase (possibly) and POD. Multiple attempts have been made to contact the manufacturers of the various systems but only Aptar replied. Descriptions of Optinose have been well published by the inventor (Djupesland) but I have included more of his information. The Kurve ViaNase has not been so well published and have not responded to my request of more information, so much of what I have obtained (and summarized in this revision) is from their website.
  3. Summary Section should be revised not to repeat your previously mentioned description in previous section. I suggest replacing this section by future directions and Expert opinion Section. Changed title of this section to discussion and added/edited accordingly.

Minor comments

Why adding a question mark in the tile? Removed

Line 26-27, rephrasing is required Rephrased.

Line 31-32, check foe meaning Rephrased

33-36, revise your description Revised

296, correct BloodBrainBarier Corrected

Line 300, correct $12 Billion by 2026[73[NasalDrug].Corrected

Line 415, correct CNS[102Mistry] Corrected

Line 439, correct PET, SPECT and gamma scintigraphy [21Trevoni] can…Corrected

Round 2

Reviewer 4 Report (New Reviewer)

In the revised version, the author appropriately addressed all the points raised by this Reviewer and, therefore, I recommend for publication this manuscript in the present version.

Reviewer 5 Report (New Reviewer)

The authors responded to all the raised comments. 

I think that the manuscript need to be English editing by an expert

This manuscript is a resubmission of an earlier submission. The following is a list of the peer review reports and author responses from that submission.

Round 1

Reviewer 1 Report

-

Reviewer 2 Report

The authors are to be commended for the clear and comprehensive overview on the upper nasal space as a target location for drug and vaccine administration. Overall, the topic has been covered at large with a proper balance between anatomical, pharmacological and basic immunological aspects to inform the readers of the manuscript. The work has been nicely illustrated with didactic figures and relevant tables. There is a particular focus on vaccine delivery and the systemic drug absorption after targeting the UNS rather than on the direct route to the brain.

I have no major remarks but only two minor suggestions:

1. Since the authors rightfully stress the importance of the medical device technology to be used for UNS drug delivery, it would be nice to add one table in which the different commercial devices are being compared in terms of mechanism, advantages and drawbacks and if relevant preferred applications. As such devices like POD, Optinose, AptarUnit etc can be compared.

2. Although it doesn't seem to be the main focus of the article, it might be interesting to mention that certain compounds, developed to specifically target brain diseases, might benefit from a lower systemic absorption or pulmonary inhalation as compared to systemic administration routes to suppress systemic adverse events and toxicity. This is especially true for substances deemed to be poorly neurotoxic but which might raise important systemic concerns ( e.g. all RNAi technology ).

Reviewer 3 Report

The manuscript reviews an interesting topic on the delivery of drugs and biologics to the upper nasal space. I feel that the topic is important, however, the manuscript in its current form cannot be accepted for publication.

1. The author has deviated from the main focus of the review and discuss so many unnecessary areas. For example, so much discussions about mucosal immunity and vaccine types are not needed. The reader for this article would like to know more about the science and technology of delivering drugs/biologics to the upper nasal space, rather than gaining knowledge on other aspects.

2. The manuscript gives too much focus to the POD system and its clinical applications. A reader would feel it like a marketing agenda rather than a scientific paper. I think emphasis should be on why conventional nasal sprays fail to deliver to the UNS and how POD and other systems overcome this. Further, the recent clinical investigations could be added, but should be briefly summarized.

3. Overall, significant rewriting and reorganization should be made to the manuscript, especially, need to add more science on nasal drug delivery and how to target UNS. Further, recent technological advancements should be added and perhaps, divided into systemic targeting and N2B targeting studies. Finally, Future goals and challenges could be included, where, vaccines and other areas can be included.      

Reviewer 4 Report

The paper reviews various drug delivery systems developed for the upper nasal space, primarily focusing on POD studies, and offers a perspective on nasal vaccine delivery. While the topic is interesting, the review's logic appears unclear. Firstly, the content of this review paper seems somewhat disconnected from the title: "The Upper Nasal Space – Why All Nasal Drug Delivery is Not the Same?" In the introduction, it states that the article will examine nasal delivery and specifically delivery to different parts of the nose. However, the subsequent content does not focus on the consequences of drug delivery to various nasal regions, why all nasal drug delivery is not the same. Instead, it primarily summarizes the devices used for nasal delivery, market trends, and the potential for nasal delivery in vaccine administration. This article appears to be more of a perspective paper on nasal vaccine delivery rather than what its title suggests. The author has published a similar topic on 2020, DOI: https://doi.org/10.17925/USN.2020.16.1.25 which better fits the title. I suggest that the author rescope the manuscript's before submission to primarily focus on the vaccine aspect.